# Current Perspectives on the Surgical Management of Perihilar Cholangiocarcinoma

**DOI:** 10.3390/cancers14092208

**Published:** 2022-04-28

**Authors:** D. Brock Hewitt, Zachary J. Brown, Timothy M. Pawlik

**Affiliations:** Department of Surgery, The Urban Meyer III and Shelley Meyer Chair for Cancer Research, The Ohio State University Wexner Medical Center, 395 W 12th Ave, Suite 670, Columbus, OH 43210, USA; daniel.hewitt@osumc.edu (D.B.H.); zachary.brown2@osumc.edu (Z.J.B.)

**Keywords:** perihilar cholangiocarcinoma, neoadjuvant therapy, liver transplantation, artificial intelligence, minimally invasive surgery

## Abstract

**Simple Summary:**

Perihilar cholangiocarcinoma is a rare disease with generally poor outcomes. Complete surgical resection remains the only chance at long-term survival. Unfortunately, most patients present with advanced, unresectable disease. Advancements in surgical technique, improved risk stratification and patient selection, and optimization of perioperative therapy have expanded the cohort of patients eligible for surgical resection. Refinement of neoadjuvant chemoradiation therapy protocols has improved outcomes after liver transplantation. Artificial intelligence to stratify patients relative to prognosis and the implementation of minimally invasive techniques, while still in the early phases of adoption and implementation, are promising areas of ongoing investigation. In this article, we discuss the current surgical management of perihilar cholangiocarcinoma.

**Abstract:**

Cholangiocarcinoma (CCA) represents nearly 15% of all primary liver cancers and 2% of all cancer-related deaths worldwide. Perihilar cholangiocarcinoma (pCCA) accounts for 50–60% of all CCA. First described in 1965, pCCAs arise between the second-order bile ducts and the insertion of the cystic duct into the common bile duct. CCA typically has an insidious onset and commonly presents with advanced, unresectable disease. Complete surgical resection is technically challenging, as tumor proximity to the structures of the central liver often necessitates an extended hepatectomy to achieve negative margins. Intraoperative frozen section can aid in assuring negative margins and complete resection. Portal lymphadenectomy provides important prognostic and staging information. In specialized centers, vascular resection and reconstruction can be performed to achieve negative margins in appropriately selected patients. In addition, minimally invasive surgical techniques (e.g., robotic surgery) are safe, feasible, and provide equivalent short-term oncologic outcomes. Neoadjuvant chemoradiation therapy followed by liver transplantation provides a potentially curative option for patients with unresectable disease. New trials are needed to investigate novel chemotherapies, immunotherapies, and targeted therapies to better control systemic disease in the adjuvant setting and, potentially, downstage disease in the neoadjuvant setting.

## 1. Introduction

Cholangiocarcinomas (CCAs) constitute a heterogenous group of epithelial cell malignancies that occur anywhere along the biliary system. These highly lethal malignancies represent approximately 15% of all primary liver cancers and 3% of gastrointestinal malignancies and cause 2% of all cancer-related deaths worldwide [1]. CCAs include three subtypes based on anatomic location along the biliary tree: intrahepatic CCA (iCCA), perihilar CCA (pCCA), and distal CCA (dCCA); pCCA represents approximately 50–60% of CCAs. The true incidence of pCCA is difficult to determine due to extensive misclassification in national databases [2]. While many CCAs arise de novo without an apparent cause, well-established risk factors include hepatitis B and C, primary sclerosing cholangitis, fibropolycystic liver disease (e.g., choledochal cysts), biliary tract stone disease, and certain genetic disorders (e.g., cystic fibrosis) [3,4]. Classically described by Klatskin in 1965, pCCAs arise between the second-order bile ducts and the insertion of the cystic duct onto the common bile duct [5,6,7]. Most pCCAs are mucin-producing adenocarcinomas with a periductal infiltrating growth pattern that eventually cause bile duct strictures and blockage [8,9]. As a result, painless jaundice is the most common presenting symptom in patients with pCCA; approximately 10% of patients present with acute cholangitis. Due to the insidious onset of disease and aggressive tumor biology, most patients have locally advanced or metastatic disease at presentation. Long-term survival is rare, with 5-year overall survival (OS) of only 6–10% for all patients with pCCA [10,11].

Complete surgical resection with negative margins provides the best opportunity for long-term survival in patients with CCA. Over the last decade, advances in multimodal therapy, regionalization of care, and aggressive surgical approaches in select patients have expanded the number of patients eligible for curative resection and improved survival [10,12,13]. Recent studies demonstrate 5-year OS following surgical resection of up to 45% [10,13,14]. We herein review the current surgical management of patients with pCCA.

## 2. Preoperative Evaluation and Optimization

### 2.1. Workup

While some patients present with non-specific symptoms such as fatigue and weight loss, pCCA often presents with painless jaundice and evidence of biliary obstruction on imaging [15]. Initial laboratory evaluation includes liver function tests and tumor markers such as carcinoembryonic antigen (CEA) and carbohydrate antigen (CA) 19-9. These tumor markers are not specific to CCA yet may provide prognostic information [16,17,18]. Furthermore, CA 19-9 can be falsely elevated in the setting of hyperbilirubinemia [19]. Novel methods to diagnose and surveil patients with pCCA include liquid biopsies, which involve the detection of markers in patient fluid samples that can be used to evaluate disease biology [20] (Figure 1). For example, Yang et al. investigated circulating tumor cells in 88 patients with CCA and reported that the presence of CTCs was associated with the extent of disease and predicted long-term survival [21].

Multiphasic, contrast-enhanced computed tomography (CT) or magnetic resonance imaging (MRI) with cholangiopancreatography (MRCP) are used to help determine resectability by characterizing the tumor and assessing the involvement of major vessels and biliary ducts, as well as identifying potential nodal or distant metastasis [23,24]. Noninvasive cholangiography with MRCP may have higher sensitivity, specificity, and diagnostic accuracy in the staging of pCCA compared with direct cholangiography (e.g., endoscopic retrograde cholangiopancreatography (ERCP)) [25]. Ideally, MRCP should be obtained prior to drainage of the biliary tree and placement of a stent to avoid artifact interference. Subsequent ERCP may be necessary to drain the biliary tree in the setting of hyper-bilirubinemia. Chest CT, with or without contrast, should also be considered to complete the staging of pCCA. Routine staging laparoscopy remains controversial. While staging laparoscopy may detect radiologically occult metastatic disease in 15–25% of patients, evaluating the presence of locally advanced disease is difficult due to the lack of tactile sensation to assess advanced biliary or vascular involvement [26]. A recent meta-analysis noted that laparoscopy benefited one in four patients; however, this rate decreased over time as cross-sectional imaging improved. Of note, approximately 50% of patients failed to undergo resection (i.e., aborted due to unresectable disease) despite a negative laparoscopy [27]. Overall, staging laparoscopy should be considered in patients with high-risk features such as large tumors, bilateral portal vein involvement, suspicious lymph nodes, and markedly elevated CA 19-9 values [14,28]. pCCA can be fludeoxyglucose (FDG)-avid. Positron emission tomography (PET)-CT may detect regional lymph nodes and distant metastases with a higher accuracy versus CT alone. Some data have suggested that the use of PET may change the management of patients with pCCA in nearly 25% of cases [29,30]. It remains unclear how much value PET-CT adds when staging laparoscopy is included with conventional staging imaging [31].

Patients with suspected pCCA should be reviewed at a multidisciplinary tumor board. Prior to any intervention, especially a biopsy, various potential therapeutic options should be considered. In particular, transperitoneal biopsy may preclude transplantation based on current protocols [32]. For patients with a high clinical suspicion of pCCA and resectable disease, biopsy may not be necessary prior to resection. If biopsy is performed, intraluminal biopsy is preferred. Among patients for whom the diagnosis of pCCA is unclear, consider serum IgG4 to rule out IgG4-related cholangitis, which may mimic CCA, as these patients do not require surgery [33,34].

### 2.2. Biliary Drainage

Preoperative biliary drainage remains an area of ongoing debate for patients with pCCA. Indications for drainage include acute cholangitis and consideration of neoadjuvant systemic therapy [35]. In a patient who has asymptomatic jaundice, the decision to perform biliary drainage should be patient-specific and discussed by a multidisciplinary team after complete staging. Liver resection performed in patients with significant hyperbilirubinemia is associated with a higher risk of postoperative complications [36,37,38]. However, manipulation of the biliary tree may cause cholangitis and sepsis that can negatively impact perioperative outcomes [39,40,41]. In addition, biliary drainage of pCCA can be complex and require multiple attempts at decompression, further increasing the risk of significant morbidity [42]. While preclinical studies suggest that hyperbilirubinemia may inhibit liver hypertrophy, clinical studies of patients with hyperbilirubinemia at the time of portal vein embolization (PVE) demonstrate no impact on liver remnant hypertrophy [43,44,45]. Overall, meta-analyses support short-duration preoperative biliary drainage in select patients with significantly elevated bilirubin levels [46,47].

When preoperative biliary drainage is needed, the optimal method of decompression remains debated. A randomized controlled trial evaluating endoscopic versus percutaneous transhepatic biliary drainage (PTBD) in patients with resectable pCCA was terminated early due to higher all-cause mortality in the percutaneous cohort [48]. However, the study had many limitations, including small sample size, concerns about the cross-over study design, and possible selection bias [49]. Compared to endoscopic methods, PTBD has lower rates of pancreatitis and cholangitis but may contribute to seeding metastasis and patient discomfort [50,51,52]. Currently, the National Comprehensive Cancer Network (NCCN) recommends endoscopic drainage as the first attempt for biliary drainage [53]. For patients requiring palliative biliary drainage in the setting of an unresectable tumor, minimally invasive approaches (e.g., endoscopic, percutaneous) are preferred over surgical approaches (e.g., cholangioenteric bypass) [54]. The specific approach, stent type (metal vs. plastic, covered vs. uncovered), and laterality (unilateral vs. bilateral) should be discussed by a multidisciplinary team.

### 2.3. Determining Resectability

The goal of curative intent surgery is to remove the lesion in its entirety with microscopic negative margins (R0 resection). A liver remnant with vascular inflow and outflow, biliary drainage, and an adequate functional liver remnant (FLR) are needed when planning the extent of resection. Retropancreatic or paraceliac lymph node involvement, extrahepatic adjacent organ invasion, or disseminated disease have all been considered relative or absolute contraindications to curative-intent surgery. Recently, high-volume centers have expanded the definition of resectability in selected patients with advanced surgical techniques, such as en-bloc resection of the portal vein or hepatic artery followed by vascular reconstruction.

Frequently used staging systems for pCCA such as the AJCC or Bismuth-Corlette classification system (Figure 2) do not predict resectability [55].

In contrast, the Blumgart staging system classifies pCCA into three clinical T stages (T1–3) and can predict resectability, likelihood of metastatic disease, and survival [56,57]. Stages are stratified by the location and extent of radial tumor growth along the biliary system, presence of portal vein involvement, and hepatic lobar atrophy [56,57]. One critique of the Blumgart staging system is that it does not take into account the more current aggressive surgical approaches often employed at major centers of excellence. While other systems exist to stratify patients by resectability status, none are routinely incorporated into clinical practice [58,59,60,61]. In general, rather than using staging or classification systems, decisions to pursue curative-intent resection should be made within the context of a multidisciplinary conference that includes experienced radiologists, medical oncologists, and surgeons.

Recently, artificial intelligence (AI) techniques have shown promise as decision-support tools to assist with diagnosis, risk stratification, and prediction of response to therapy [62]. Radiomics quantifies textural information from images, including spatial distribution of signal intensities and pixel interrelationships, using AI methodologies such as machine learning. Radiomic signatures have been reported to perform well (AUC ≥ 0.90) in models to predict lymph node metastasis in patients with CCA [63,64]. While prognostic tools utilizing radiomic evaluation require validation in prospective studies, AI analytics have the potential to provide actionable information and assist with clinical decision making by integrating vast quantities of data from multiple sources (e.g., clinical, imaging, genomics, molecular, etc.) to inform patient-specific therapy plans.

### 2.4. Future Liver Remnant

Postoperative hepatic insufficiency represents a major source of potential serious morbidity and mortality after major hepatectomy. Optimizing the FLR provides protection against the risk of liver insufficiency or failure. An adequate FLR requires at least two continuous segments with adequate venous and arterial inflow, venous outflow, and biliary drainage. For patients with a healthy non-diseased liver, the FLR threshold is generally 20–30% [65]. The extent of underlying liver disease from steatohepatitis, cirrhosis, chemotherapy-associated liver injury, or other hepatotoxic sources influences the amount of FLR needed to mitigate the risk of post-hepatectomy complications. The NCCN guidelines recommend at least 30–40% liver remnant in patients with Child-Pugh Class A cirrhosis [35]. In situations where the FLR size may be questionable, CT or MRI volumetry and/or liver function assessment with scintigraphy or indocyanine green (ICG) clearance should be performed [66,67,68]. If volumetry demonstrates an inadequate FLR, liver hypertrophy-inducing interventions are required. Two interventions to address an inadequate FLR are PVE and associating liver partition and portal vein ligation (ALPPS). The concept of blocking the portal vein to induce hypertrophy of the non-occluded liver was first introduced over 100 years ago in an animal experiment [69]. It took over 50 years, however, for the first clinical application of portal vein occlusion by Honjo et al. [70]. Over the subsequent two decades, the technique of percutaneous trans-hepatic PVE was refined, and today, it is safely performed to increase FLR volume prior to hepatectomy. In appropriately selected patients, PVE induces liver hypertrophy, leading to higher utilization of hepatectomy with lower rates of postoperative hepatic insufficiency [71]. In situations where PVE does not generate sufficient hypertrophy, sequential hepatic vein embolization can safely stimulate additional FLR hypertrophy [72]. However, simultaneous portal and hepatic vein embolization may cause higher rates of liver remnant portal vein thrombosis without additional hypertrophy over PVE alone [72]. HYPER-LIVo1 is an active randomized controlled phase 2 trial currently investigating the efficacy of simultaneous portal and hepatic vein embolization [73].

Based on the concept of a two-stage hepatectomy, which was first described to treat bilateral colorectal liver metastases over two decades ago, the ALPPS procedure was proposed by Dr. Hans Schlitt in 2012 [74]. The first stage combines portal vein ligation of the diseased liver with liver parenchymal transection along the FLR. Adequate liver hypertrophy generally occurs in 1–2 weeks. In the second stage, the surgeon completes the hepatectomy by transecting the vascular inflow and outflow as well as the biliary duct [75]. ALPPS induced greater liver hypertrophy, and patients had a higher rate of stage 2 hepatectomy completion compared with historical two-stage hepatectomy; however, high morbidity and mortality have prevented broader adoption of ALPPS [76,77]. Partial ALPPS, where the portal vein is similarly ligated, but only a portion of the liver parenchyma is transected, first described by Alverez et al. in 2015, may provide similar liver hypertrophy with significantly less morbidity and mortality, especially when performed with a minimally invasive approach [78,79,80]. Evaluation of historical studies comparing ALPPS and PVE generally shows that ALPPS produces greater FLR hypertrophy with higher rates of two-stage hepatectomy, but PVE has lower morbidity and mortality [77,81]. In fact, while endorsed for the treatment of colorectal liver metastasis, the high incidence of morbidity and mortality associated with the use of ALPPS for pCCA has resulted in general avoidance of this approach.

## 3. Surgical Resection

Complete surgical resection with microscopically negative margins (R0) remains the best change at long-term survival for patients with pCCA. Unfortunately, less than 40% of patients present with resectable disease. In addition, nearly 50% of these patients have unresectable disease at surgical exploration [14]. Among patients who undergo an R0 resection, 5-year OS can reach 45%, which is much higher than the 0–23% 5-year OS reported for patients who undergo a margin positive resection (R1 or R2) [56,82,83,84,85].

Standard resection of a pCCA generally involves either a right extended hepatectomy or left hemi-hepatectomy with concomitant bile duct resection and porta hepatis lymphadenectomy with bilioenteric reconstruction. Caudate resection improves the incidence of margin negative resection and survival without significant additional morbidity and should be included in the resection [86]. Bilioenteric anastomosis is commonly performed with a Roux-en-Y hepaticojejunostomy. While mucosa-to-mucosa apposition is standard, small bile ducts may be encountered that make such an anastomosis difficult. In these situations, anastomosis without mucosa-to-mucosa alignment has acceptable long-term outcomes [87].

In some instances, hepatectomy needs to also involve vascular resection to achieve negative margins. Vascular resection can involve the hepatic artery alone, the portal vein alone, or combined resections with reconstruction of the liver remnant. Left-sided vascular resection and reconstructions are generally less common and less technically demanding as the left portal vein has a long extrahepatic course with easier access to the vein in the umbilical fissure. In addition, the left hepatic artery is infrequently involved with tumor as it runs away from the biliary confluence. Conversely, the right portal vein bifurcates early, and the right branches often have significant size discrepancies with the main portal vein [88]. Given that pCCA arises at the biliary confluence, it is not uncommon that the tumor involves the right hepatic artery as it courses behind the common hepatic duct. Traditionally, patients requiring a vascular construction had worse outcomes with unacceptable morbidity and mortality compared with patients who did not undergo vascular reconstruction [89,90]. Improvements in patient selection, surgical technique, and perioperative management have led to acceptable morbidity and mortality after vascular resection and reconstruction at high-volume, specialized centers. In two recent large cohort studies, perioperative morbidity and mortality were similar among patients who underwent hepatectomy with versus without vascular resection [91,92]. Of note, median OS (30–36 months vs. 45–61 months) was shorter among patients who underwent vascular resection compared to patients that did not receive a vascular resection, yet longer than patients who did not undergo resection at all (10 months). Operations that involve a major hepatectomy with vascular reconstruction can be technically very challenging and should only be performed at specialized centers.

The periductal growth pattern of pCCA can result in high rates of microscopically positive margins (R1). In turn, intraoperative frozen section of the proximal and distal ductal margin is sometimes performed to evaluate the completeness of the resection. If the frozen section is positive, the surgeon should re-resect the margin to achieve a negative margin if further resection can be performed without major morbidity [35]. Patients with an R0 resection after re-resection of a positive frozen section margin have similar survival to patients with an upfront R0 resection; however, patients with a persistent R1 margin have worse survival [93]. Additional resection of the bile duct can be technically challenging, especially as disease approaches the second-order bile ducts proximally or may require a pancreaticoduodenectomy distally.

For patients with limited or mid bile duct tumors (i.e., Bismuth-Corlette type 1), limited bile duct resection with frozen section assessment of the distal and proximal margins has been attempted in select cases. En-bloc resection with a hepatectomy or pancreaticoduodenectomy provides a higher likelihood of an R0 resection and improved survival compared to bile duct resection alone and, therefore, is the preferred approach [56,94]. Conversely, for tumors with extensive bile duct involvement, a combined pancreaticoduodenectomy and hepatectomy may be needed to achieve negative margins. Mortality associated with combined pancreaticoduodenectomy and hepatectomy may be much higher at approximately 10%. Patients undergoing combined pancreaticoduodenectomy and hepatectomy who achieved an R0 resection have reported 5-year survival ranging from 18 to 68%, while no patients with positive margins survived to 5 years [95].

### 3.1. Minimally Invasive Surgery

Over the last decade, minimally invasive surgical (MIS) approaches to pCCA have increased in centers across the world, hastened by the broader adoption of robotic technology. Most published reports are small, single-institution retrospective cohort studies from facilities outside of the United States that address safety and feasibility [96,97,98,99]. Few studies provide direct comparisons of patient outcomes between MIS and open surgery. As such, studies that include both approaches are heavily influenced by selection bias. Overall, patients who underwent MIS have less reported postoperative pain and shorter hospital lengths of stay with comparable perioperative complications and 1-year mortality with a pooled conversion rate of 5.5% [96,97,99]. The robotic approach may provide benefit for difficult bilioenteric anastomoses. MIS approaches can achieve adequate lymphadenectomy and nodal evaluation; however, the incidence of caudate lobectomy in MIS cases has been much lower, suggesting that this procedure may be more challenging with a non-open approach [98]. While data on long-term oncologic outcomes following MIS for pCCA are still emerging, the lower rates of caudate resection may result in higher local recurrence rates [98]. In addition, data regarding MIS vascular resection and reconstruction are limited for patients with pCCA. While MIS approaches are safe with acceptable short-term morbidity and mortality, the approach should be limited to surgeons and centers with MIS expertise.

### 3.2. Liver Transplantation

Orthotopic liver transplant for pCCA had been contraindicated based on early reports that demonstrated high rates of tumor recurrence with transplant [100,101]. Over the last two decades, data emerged demonstrating that highly selected patients with pCCA benefited from transplantation after a neoadjuvant protocol that included neoadjuvant chemoradiation (Mayo Protocol) [102,103,104]. Specifically, patients with early-stage (≤3 cm) unresectable pCCA, as well as individuals with disease arising in the setting of primary sclerosing cholangitis, may be candidates for transplantation. These individuals are typically treated with a combination of external beam radiotherapy, radio-sensitizing chemotherapy, brachytherapy, and/or maintenance chemotherapy until liver transplant [105] (Figure 3). 

Results from a multicenter study that implemented a neoadjuvant chemoradiation protocol prior to transplantation demonstrated an intention-to-treat survival rate at 2 and 5 years of 68% and 53%, respectively, and recurrence-free survival of 78% and 65%, respectively [105]. Several factors were associated with worse outcomes: previous trans-peritoneal biopsy, presence of metastatic disease, or history of other malignancy. Of note, approximately 1 in 10 patients dropped out of the neoadjuvant protocol prior to liver transplant. A recent meta-analysis confirmed the survival benefit of transplantation for well-selected patients with pCCA treated with neoadjuvant chemoradiation therapy and liver transplant [106]. Specifically, patients treated with transplantation compared with resection had an improved OS at 3 years (72% vs. 33%) and 5 years (64% vs. 18%) [107]. Transplant remained associated with improved survival even on the intention-to-treat analysis after adjusting for tumor size, lymph node status, and primary sclerosing cholangitis. The TRANSPHIL trial, a randomized, prospective, multicenter phase III study comparing resection to neoadjuvant chemoradiotherapy and liver transplant for patients with resectable pCCA is currently accruing (NCT02232932) (Table 1). Patients who meet transplant criteria with unresectable pCCA should be strongly considered for transplantation and discussed at a multidisciplinary conference that includes transplant surgeons. Among patients with PSC-related pCCA, transplantation should even be considered among individuals with resectable disease, given the improved outcomes for this subset of patients following transplantation.

### 3.3. Neoadjuvant Systemic Therapy and Radiation

Proponents of neoadjuvant therapy cite the opportunity to treat occult metastases, increase R0 resection rates, and determine which patients are most likely to benefit from resection (i.e., if disease progression on therapy, then individuals are less likely to benefit from surgery). A recent National Cancer Database study noted that CCA patients who received neoadjuvant chemotherapy had longer median OS versus patients who received adjuvant chemotherapy alone (40.3 months vs. 32.8 months, *p* = 0.01) [109]. Despite the fact that the authors used propensity score matching, these data need to be considered cautiously given the retrospective nature of the study. In a single-institution study, neoadjuvant therapy with gemcitabine and S-1 was evaluated among patients with resectable, borderline resectable, and locally advanced pCCA. While safe, with most patients able to complete treatment (91%), neoadjuvant therapy was not associated with disease-specific survival [59]. A systematic review on neoadjuvant therapy concluded that there was some potential benefit to downsizing the lesion to improve the likelihood of an R0 margin, yet further prospective studies were needed to determine the role and benefit of preoperative therapy for patients with pCCA [110].

Over the last decade, there have been significant advances related to targeted therapies and immunotherapies for patients with cancer. Unfortunately, the role of immunotherapy and targeted therapy for CCA has lagged behind many other cancers [1]. In 2021, the FDA did approve ivosidenib for patients with locally advanced or metastatic CCA with an isocitrate dehydrogenase-1 mutation. Unfortunately, <1% of extrahepatic CCA have an IDH1 mutation versus 13% of iCCAs [111]. The FDA has also approved pembrolizumab for patients with unresectable or metastatic microsatellite instability-high or mismatch repair deficient solid tumors (including CCA) that progressed on prior treatment. In an analysis of only patients with advanced biliary tract disease from KEYNOTE-158 and KEYNOTE-028, the results were underwhelming; pembrolizumab provided an objective response rate of 6–13% and a median progression-free survival of ≤2 months [112]. No data on the use of immune or targeted therapies in the neoadjuvant setting for patients with pCCA have been published to date. Future therapeutic approaches to patients with pCCA will most likely include combination therapy targeted to patients who are most likely to benefit based on molecular profiling of the cancer tissue [113].

## 4. Conclusions

Complete surgical resection remains the best chance at potentially curative-intent treatment of patients with pCCA. Advanced surgical approaches and adjuncts, including vascular resection and reconstruction, PVE, and neoadjuvant chemoradiation with transplant, have expanded the therapeutic options for patients with pCCA. Minimally invasive techniques and AI technologies, while in the early stages of adoption and implementation, have shown promise to enhance the management of patients with pCCA. Ongoing investigations into novel biomarkers, including liquid biopsy, and perioperative systemic therapies (e.g., chemotherapy, immunotherapy, and targeted therapy) are needed to improve early disease detection, risk stratification, surgical management, and long-term outcomes of patients with pCCA.

## Figures and Tables

**Figure 1 cancers-14-02208-f001:**
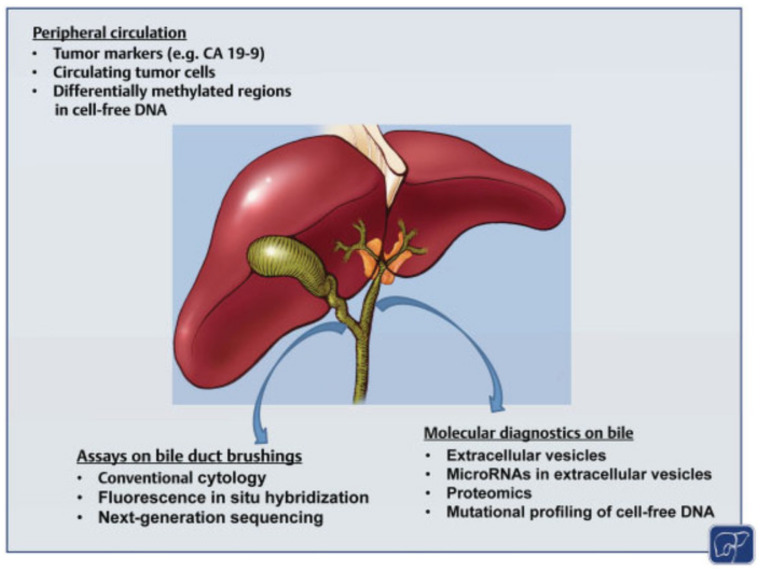
Biomarkers for perihilar cholangiocarcinoma [22].

**Figure 2 cancers-14-02208-f002:**
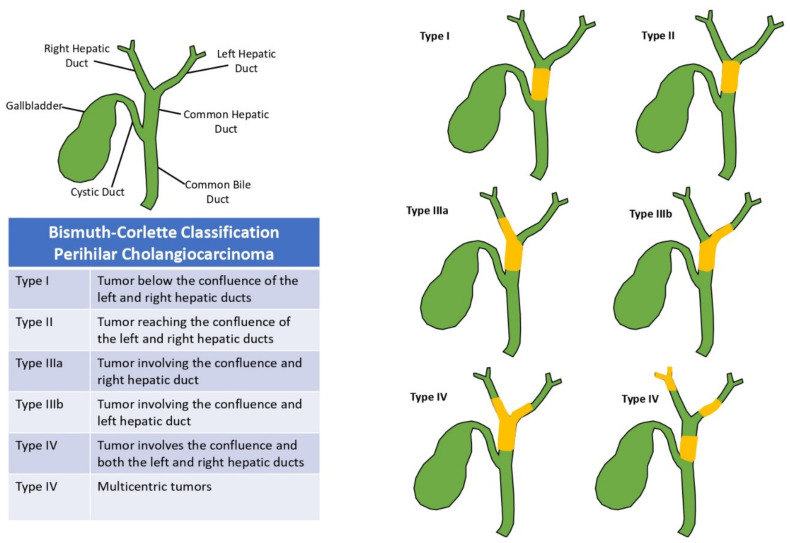
Bismuth-Corlette classification for perihilar cholangiocarcinoma.

**Figure 3 cancers-14-02208-f003:**
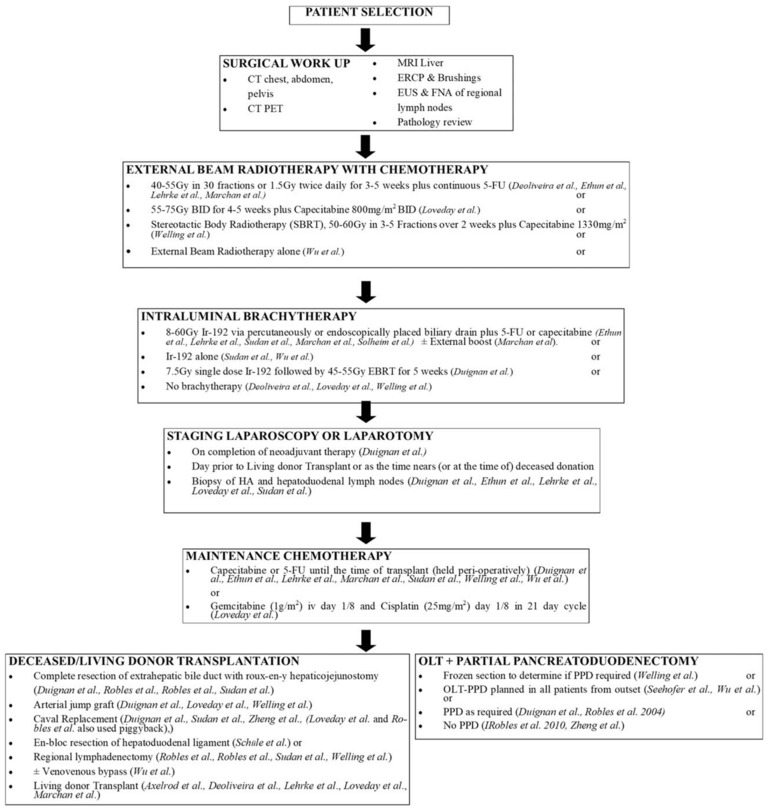
Neoadjuvant protocols leading to liver transplant in perihilar cholangiocarcinoma [106].

**Table 1 cancers-14-02208-t001:** Neoadjuvant trials including perihilar cholangiocarcinoma.

Trial ID/Name	Location	Trial Type	Tumor Site	Resectability Status	No. of Patients	Intervention	Primary Outcomes	Status
NCT02232932	France	Phase III	pCCA	Resectable	60	Capecitabine-radiotherapy-liver transplant v resection	OS	Active, not recruiting
NCT03673072 (GAIN)	Germany	Phase III	GBC, CCA	Incidental diagnosis post cholecystectomy or advanced CCA	300	Cisplatin + gemcitabine (×3 cycles) v nil → surgery → +/- adjuvant cisplatin + gemcitabine (×3 cycles)	OS	Recruiting
NCT03603834	Thailand	Phase II	CCA	Borderline resectable	25	mFOLFOXIRI	ORR	Recruiting
NCT04308174 (DEBATE)	Korea	Phase II	GBC, CCA	Resectable	45	Durvalumab + cisplatin + gemcitabine v cisplatin + gemcitabine	R0 rate	Recruiting
NCT04727541	Germany	Phase II	GBC, CCA	Resectable	24	Bintrafusp-alfa ×2 doses	Major pathologic response	Recruiting
NCT04480190	USA	Phase I	GBC, CCA	Resectable	12	Gemcitabine + cisplatin + 5-FU/RT	Completion of therapy	Recruiting
NCT04378023	Spain	Phase IV	pCCA	Unresectable	34	EBRT + capecitabine → cisplatin + gemcitabine until transplant	OS at 1, 3, and 5 years	Recruiting
NCT04824742	China	Phase II	CCA	Resectable	50	PDT	R0, local recurrence, OS 5-year	Not yet recruiting

As per clinicaltrials.gov on 30 March 2022. CCA—cholangiocarcinoma; GBC—gallbladder carcinoma; pCCA—perihilar cholangiocarcinoma; OS—overall survival; EBRT—external-beam radiotherapy; mFOLFOXIRI—fluorouracil + oxaliplatin + irinotecan; ORR—overall response rate; PDT—photodynamic therapy. Modified and adapted from [108].

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
