# Peer review of "Current Perspectives on the Surgical Management of Perihilar Cholangiocarcinoma"

_cancers, 2022, doi:10.3390/cancers14092208_

Round 1

Reviewer 1 Report

This update on perihilar cancer is well written, extensive and perfectly referenced.
It is an excellent base for those who want to have up-to-date knowledge on the subject. I recommended "accept after minor revisions" and my concerns are  :

- Figure 1 could be replaced by a table. It would be easy then to explain in one sentence each of the reported explorations.

- Page 4 para 2.2 line 4. "perform" instead of "performed"

- Page 4 para 2.3 last sentence. Please provide the reader with a reference.

- About para 2.3 : There is often confusion between staging and resectability.
Staging has a prognostic value and is only accurately done postoperatively
on the specimen. This must be mentioned. The BC or SKMCC classifications assess resectability.
Also I found that authors are too quick to sweep aside the BC classification and give pride of place to Blumgart's classification. To my knowledge, very few teams use the Blumgart classification. Could you temper your eagerness to make this classification the gold standard ? This is especially true since authors reffered to the Bismuth classification in the last paragraph on page 7 (BC type 1) 
Page 6 para 2 line 7. A reference is needed. I suggest this one: Boudjema et al. J Gastrointest Surg. 2013 Jul;17(7):1247-56. 

Author Response

Thank you for reviewing our manuscript entitled “Current Perspectives on the Surgical Management of Perihilar Cholangiocarcinoma.”

We are pleased that the manuscript was favorably reviewed and was found to be potentially acceptable for publication pending revisions.

We thank the reviewers for the valuable insight and comments as these serve to further strengthen our manuscript.

As requested, we have provided a point-by-point response to each of the reviewers’ comments with relevant changes made to the manuscript.

Reviewer 1
1. Figure 1 could be replaced by a table. It would be easy then to explain in one sentence each of the reported explorations.

            We thank you for your constructive comments. There are certainly benefits to displaying the information on biomarkers in a table format with additional explanation; however, we believe the figure best demonstrates the variety of biomarkers under investigation without going into too much detail, which would be beyond the scope of the current article.

  1. Page 4 para 2.2 line 4. "perform" instead of "performed"

This correction has been made.

  1. Page 4 para 2.3 last sentence. Please provide the reader with a reference.

            References for this sentence include the two listed below.

  1. Mizuno T, Ebata T, Yokoyama Y, et al. Combined Vascular Resection for Locally Advanced Perihilar Cholangiocarcinoma. Ann Surg. 02 01 2022;275(2):382-390. doi:10.1097/SLA.0000000000004322
  2. Sugiura T, Uesaka K, Okamura Y, et al. Major hepatectomy with combined vascular resection for perihilar cholangiocarcinoma. BJS Open. 07 06 2021;5(4)doi:10.1093/bjsopen/zrab064
  3. The reviewer had concern regarding the verbiage on staging vs resectability as well as promoting Blumgart over Bismuth-Corlette. In particular, “- About para 2.3: There is often confusion between staging and resectability. Staging has a prognostic value and is only accurately done postoperatively on the specimen. This must be mentioned. The BC or SKMCC classifications assess resectability. Also I found that authors are too quick to sweep aside the BC classification and give pride of place to Blumgart's classification. To my knowledge, very few teams use the Blumgart classification. Could you temper your eagerness to make this classification the gold standard ? This is especially true since authors reffered to the Bismuth classification in the last paragraph on page 7 (BC type 1) “

            We thank the Reviewer for this comment. As requested, we have added clarification on the topic as highlighted in the manuscript on pages 4 and 5. We agree that staging systems should have prognostic value and pathologic staging provides strong prognostic value. There is also, however, significant prognostic value in clinical staging. In particular, studies that have evaluated the Blumgart system, and other clinical systems, have demonstrated strong evidence that these approaches can stratify patients relative to survival. In the current paper, we choose to highlight the Blumgart clinical T stage system as it has clinical applicability in that it takes into account vascular involvement, a significant factor determining resectability. Ultimately, there is not one staging or classification system that has been adopted or embraced by all clinicians. We have stressed this point at the end of the paragraph by mentioning that no system is routinely incorporated into clinical practice and that management decisions should be made within the context of a multidisciplinary setting.

  1. Page 6 para 2 line 7. A reference is needed. I suggest this one: Boudjema et al. J Gastrointest Surg. 2013 Jul;17(7):1247-56.

            As requested, the reference was added.

Reviewer 2 Report

The manuscript represents a narrative review about perihilar cholangiocarcinoma. 

In figure 1, as example for markers, I recommend authors to add CEA. 

Some palliative procedures (transtumoral drilling, stenting), should be detailed. 

Klatskin tumour classification Bismuth-Corlette should be added with a suggestive figure.

Author Response

  1. In figure 1, as example for markers, I recommend authors to add CEA.

            Thank you for your constructive comments and suggested revisions. The image is used unaltered with permission from the original authors/publisher. We mentioned both CEA and CA 19-9 in the first paragraph under Workup.

  1. Some palliative procedures (transtumoral drilling, stenting) should be detailed.

            As requested, discussion on palliative procedures was added on page 4.

  1. The reviewer suggested adding a figure on the Bismuth-Corlette classification.

As requested, we added figure 2 to depict the Bismuth-Corlette classification (page 4).

Thank you for considering our revised manuscript.

Sincerely,

Timothy M. Pawlik, MD, MPH, MTS, PhD, FACS, FRACS (Hon.)

Professor and Chair, Department of Surgery

The Urban Meyer III and Shelley Meyer Chair for Cancer Research

Professor of Surgery, Oncology, and Health Services Management and Policy

Surgeon in Chief, The Ohio State University Wexner Medical Center

The Ohio State University, Wexner Medical Center